# Serum Cortisol and Its Correlation with Leucocyte Profile and Circulating Lipids in Donkeys (*Equus asinus*)

**DOI:** 10.3390/ani12070841

**Published:** 2022-03-26

**Authors:** Daniela Alberghina, Alessandra Statelli, Vincenzo Monteverde, Irene Vazzana, Giuseppe Cascone, Michele Panzera

**Affiliations:** 1Dipartimento di Scienze Veterinarie, Università degli Studi di Messina, Via Palatucci sn, 98168 Messina, Italy; alessandra.statelli@unime.it (A.S.); mpanzera@unime.it (M.P.); 2Experimental Zooprophylactic Institute of Sicily, Via Gino Marinuzzi 3, 90129 Palermo, Italy; vincenzo.monteverde@izssicilia.it (V.M.); irene.vazzana@izssicilia.it (I.V.); 3Experimental Zooprophylactic Institute of Ragusa, Contrada Conservatore, 97100 Ragusa, Italy; giuseppe.cascone60@gmail.com; 4Centro Universitario Specializzato per gli Interventi Assistiti con gli Animali, Università degli Studi di Messina, 98168 Messina, Italy

**Keywords:** donkeys, cortisol, leucocyte profile, lipids

## Abstract

**Simple Summary:**

There are relatively few studies investigating serum cortisol levels in donkeys. The aim of this study was to evaluate basal cortisol values in donkeys, and its physiological role on the immune system and lipid levels in relation to age and pregnancy. The results showed that age did not influence cortisol levels while pregnancy increased its concentrations. Different correlations were found between cortisol, leucocyte types, and serum triglycerides depending upon age and pregnancy state. Although correlation is, of course, not equivalent to causation, we can hypothesize that the multiple functions of cortisol on the immune system and lipid metabolism vary across donkey lifespan and are related to physiological state.

**Abstract:**

The values for basal serum cortisol concentrations of horses are available in many studies. However, there are limited data about serum cortisol in donkeys. The present study aimed to determine the baseline values for serum cortisol, to evaluate the influence of age and pregnancy on its levels, and to correlate its values with leucocyte profile, serum cholesterol, and triglycerides. Serum samples were collected from 97 healthy donkeys. Cortisol was analyzed by chemo-luminescent assay. The median and the 2.5th and 97.5th percentiles of serum cortisol measured and calculated in all donkeys were 5.64, 3.40, and 10.54 µg/dL, respectively. Females (n.91) were divided into three groups: Group A (young), Group B (adult), and Group C (pregnant at the 9th–11th months). The effect of age and physiological status was investigated by the Mann–Whitney test. Group C showed significantly higher levels than Group B (*p* < 0.05). Significant correlations were found in Group B with monocytes (r = 0.37, *p* < 0.01) and triglycerides (r = 0.30, *p* < 0.05), and in Group C with monocytes (r = 0.79, *p* < 0.01), basophils (r = 0.6, *p* < 0.05), and neutrophil/lymphocyte ratio (r = −0.63, *p* < 0.05). Higher cortisol values related to late pregnancy are also found in this species. These preliminary results provide evidence for a relationship between cortisol and the immune system as well as cortisol and lipid metabolism modulated by age and pregnancy when parameters are within normal values.

## 1. Introduction

Cortisol is a steroid hormone produced and secreted by the zona fasciculata layer of the adrenal cortex and is the final product of the hypothalamic–pituitary–adrenal (HPA) axis. The increased HPA axis response is one of the most reported physiological responses to animal stress [1]. Cortisol plays a primary role in gluconeogenesis, proteolysis, lipolysis, hyperglycemia, modulation of immunity and inflammation [2], sexual development, and reproductive function [3]. Cortisol also regulates bone metabolism [4]. Cortisol has positive effects on traits related to robustness and adaptation: newborn survival was shown to be directly related to plasma cortisol levels at birth, and resistance to bacteria and parasites is increased in animals selected for a higher HPA axis response to stress [5]. Furthermore, the tolerance to heat stress is better in individual animals able to mount a strong stress response [5]. Cortisol is necessary for the maintenance of memory [6] and facilitates effective learning via the functioning of the hippocampus [7]. The facilitation of learning is clearly a positive function, as is the increase in cortisol during courtship, mating, and the exertion to obtain food [8]. Memory and learning appear to be important factors influencing animal survival and fitness [6]. It is evident that cortisol is not a “bad guy,’’ but rather it has a normal physiological role, which not only helps animals, including humans, adapt to stressors but also coordinates our metabolism with our daily activity and sleep patterns [9,10].

Appropriate veterinary care for donkeys is challenging despite being important working animals in non-industrialized countries and pets in first world countries [11]. In recent years, the growing use of donkey milk, especially in children allergic to cow’s milk proteins [12], and their use in Animal-Assisted Interventions [13], has seen an increasing number of donkey farms throughout Italy. Donkeys have a unique physiology when compared with the horse, and a different set of reference ranges should be used when assessing the donkey patient [14]. The high variability in cortisol concentrations in horses (*Equus caballus*) has led to the determination of reference values according to physiological status, age, management, and environmental conditions in different breeds [15,16]. Despite extensive cortisol studies in horses [15,17], there are limited data related to serum cortisol levels in donkeys [18]. Donkeys are very resistant to stress [19], are more stoical in their behaviour, and tend to agitate less readily than horses. Scientific literature shows that asinine cortisol increased after transport [20,21,22] and during the estrous cycle [23]. Food deprivation increased cortisol and triglyceride concentrations in the serum of adult donkeys [20]. Hyperlipemia is a common disorder of the donkey [24], and stress is a risk factor for this disease [25]. Since serum cortisol, as a steroid, is derived from cholesterol, and lipids are usually metabolized to release energy under stress, there may be a possibility that baseline levels of cortisol could be associated with baseline levels of cholesterol and triglycerides in relation to age or physiological conditions. To the author’s knowledge, no research has been conducted to assess basal concentrations of cortisol and to explore its interactions with leucocytes and serum lipids in donkeys.

The aim of this study was threefold: (1) to evaluate basal values for cortisol in donkeys, (2) to examine the influence of age and pregnancy on its levels, and (3) to test a possible association between cortisol and leucocyte profile and serum lipids.

## 2. Materials and Methods

The protocol of animal husbandry and experimentation was reviewed and approved in accordance with the standards recommended by the Guide for the care and use of Laboratory Animals and Directive 2010/63/EU for animal experiments. According to the mentioned legal regulations, the study cannot legally be classified as an animal experiment, as blood sampling is a common part of health monitoring in donkeys in accordance with good veterinary practice. The further use of samples obtained during clinical veterinary procedures for research purposes is not considered an animal experiment, so no ethical permit is required according to Italian legislation. 

### 2.1. Animals

After a complete clinical exam, 97 clinically healthy Ragusana (n. 77) and Sarda (n. 20) donkeys (91 females and 6 males) from n. 2 breeding farms located in Ragusa (Sicily, 36°55′45″48 N 14°43′4″80 E), were enrolled in this study. Animals were healthy during the visit, based on their recorded history and their clinical inspection. Since they were familiar with humans, they were not restrained during blood collection. Morning blood samples (9.00–11.00 a.m.) were collected in February 2020. In both milk production farms, all donkeys lived in small paddocks and were fed grass supplemented with hay and/or concentrate. The daily ration included a single distribution of hay (5.0 kg/animal/day) and commercial concentrate feed (2.5 kg/animal/day), and they were free to graze, while water was available ad libitum. Animals had no access to concentrate feed the night before testing. The number of donkeys per farm was 35 and 62, with a similar ratio of female/male, barren/pregnant females, and breeds. Female animals were divided into three groups according to their age and physiological state (pregnancy): Group A (9 donkeys from 1 to 3 years old), Group B (70 animals from 5 to 22 years old), and Group C (n. 12 pregnant jennies, age from 4 to 9 years old).

### 2.2. Blood Sample Collection and Analysis

Blood samples from each animal were collected by means of a jugular venipuncture into a 10 mL vacutainer tube containing EDTA for determination of haematological values and into a 10 mL tube with clot activator (Terumo Corporation, Tokyo, Japan) for serum measurement of cortisol. Blood samples were placed in refrigerated bags and transported to the laboratory for analysis. EDTA whole blood samples were processed in the laboratory within 2 h. Haematological and cortisol analyses were carried out in the laboratory of the Istituto Zooprofilattico Sperimentale of Palermo. Total leukocyte (WBC), neutrophils (NEU), lymphocytes (LYM), monocytes (MON), eosinophils (EOS), and basophils (BAS) were measured by means of an automated hematology analyzer (Cell-Dyn 3700 Abbott GmbH & Co.KG Wiesbaden, Germany). Neutrophil-to-lymphocyte ratio (N/L) was calculated. Samples with clot activator were allowed to clot for 30 min at room temperature and thereafter centrifuged. Serum was divided into aliquots and stored at −20 °C until cortisol and lipid analysis (within one month from blood collection). Serum cortisol concentration was measured with the IMMULITE 2000 (Siemens Healthcare Diagnostic, Deerfield, IL, USA). The serum concentration of cholesterol (CHOL) and triglycerides (TRY) was assessed using commercially available kits by means of an automated analyzer (Boehringer Mannheim/HITACHI 911, Roche, Basel, Switzerland).

### 2.3. Statistical Analysis

Data analysis was performed using GraphPad prism version 8.0 statistical software (GraphPad Software Ltd., San Diego, CA, USA). Cortisol concentration was tested for normality by applying the Shapiro–Wilk’s test (*p* > 0.05). Since the resulting data were not normally distributed, the values were expressed as median and 2.5th and 97.5th percentiles [22]. A Mann–Whitney test was conducted to determine how the categories differed. Pearson’s correlation coefficients were computed to evaluate the relationship between serum cortisol concentration and age, serum cortisol concentration and leucocyte profile, and serum cortisol concentration and lipids. A linear regression model (y = a + bx) was applied to determine the degree of correlations between parameters. The level of statistical significance was set at *p* < 0.05.

## 3. Results

### 3.1. Basal Cortisol and Influence of Age and Pregnancy

Donkeys included in this study had a mean age of 7.7 (±3.6) and an age range between 1 and 22 years. The median and the 2.5th and 97.5th percentiles of serum cortisol measured and calculated in all donkeys was 5.64, 3.40, and 10.54 µ/dL, respectively. No differences among age groups were found, but levels of cortisol were significantly higher in the pregnant mares compared to adults *p* < 0.05 (Table 1).

### 3.2. Correlations of Cortisol with Leucocytes and N/L Ratio

Leucocyte values in all donkeys and broken down by group of females are shown in Table 2. Compared to Group B, Group A showed significantly lower EOS (*p* < 0.001) and Group C showed significantly lower EOS and BAS (*p* < 0.05 and *p* < 0.01, respectively).

As shown in Figure 1, in Group B, a positive correlation was found between cortisol and monocytes (r = 0.37, *p* < 0.01) and in Group C, a positive significant correlation was found between cortisol and monocytes (r = 0.79, *p* < 0.01), basophils (r = 0.6, *p* < 0.05), and a negative significant correlation with N/L ratio (r = −0.63, *p* < 0.05).

### 3.3. Correlations of Cortisol with Serum Cholesterol and Tryglicerids

Serum TRI and CHOL concentrations in all donkeys and broken down by group of females are shown in Table 3. The obtained values were within ranges [26]. Compared to Group B, Group A showed significantly higher CHOL levels (*p* < 0.01) and significantly lower TRI levels (*p* < 0.05).

As shown in Figure 2, in Group B, a weak but significant positive correlation was found between cortisol and TRI (r = 0.30, *p* < 0.05).

## 4. Discussion

To the best of our knowledge, this is the first study to have evaluated basal serum cortisol in donkeys. To establish population-based reference intervals, blood should be sampled from a large number of animals of a particular species, using a mixture of breeds, ages, and sex. Unfortunately, the examined donkey population in our study consisted of mostly females (88%). Basal cortisol concentrations in healthy adult horses at rest are reported in the approximate range of 1.1 to 14.3 μg/dL (30 to 395 nmol/L) [26,27]. Observed cortisol levels (3.40–10.54 μg/dL) seem to overlap with the normal range for horses, and they were slightly higher than those reported by Dugat et al. [18] (2.4–6.0 μg/dL as 5–95% ranges). Due to the small sample size of Dugat’s study (*n* = 44), differences related to age or physiological state (pregnancy) were not evaluated; furthermore, no information on laboratory methods was given. Our results were obtained during winter, whereas in the previous study, data were obtained in late spring. Plasma adrenocorticotropic hormone concentrations show seasonal changes in both horses and donkeys [28], but cortisol concentrations do not vary in horses [29]. It is possible that cortisol does not show seasonality in donkeys either, but a specific investigation with the same animals tested during seasonal changes has not yet been conducted for this species. Values of adult jennies in our study were compared to values of young and pregnant jennies. Cortisol of pregnant jennies was significantly higher. The HPA axis is activated during normal pregnancy, resulting in a physiologic hypercortisolism [30]. The higher cortisol levels in the 9th-11th month of pregnancy could be related to fetus production at the end of the pregnancy, and confirms results previously reported in this species [31]. Glucocorticoids are naturally increased at the end of pregnancy for fetal brain and lung development [32] and act as major mediators of leukocyte distribution [33]. Obtained values of leucocyte profiles were within ranges [26]. We tested the hypothesis that there may be a correlation between cortisol and leucocyte count and/or N/L ratio in healthy donkeys. In general, cortisol increases production and circulating levels of leukocytes and neutrophils in particular. Compared to adult females, eosinophils were significantly lower in young and pregnant jennies. The number of eosinophils tends to increment with age, probably due to the progressive exposure of animals to parasites during their life [34]. The relationship between glucocorticoid concentration and eosinopenia is well-established, being supported by many experimental and clinical studies [35,36]. It is possible that similar mechanisms of cortisol control on eosinophils exist during pregnancy. Cortisol levels were significantly positively correlated with monocytes in adults and in pregnant jennies. Corticosteroids, exogenous and increased endogenous secretion, cause monocytosis and eosinopenia. Horses usually show no change in number of monocytes as an effect of corticosteroids [37]. No difference was found between adults and pregnant jennies in number of monocytes. In rodents, it has been shown that cortisol decreases circulating monocyte levels [38]. The discrepancy among species could be explained by the severity of the stressor or potentially by sex. The correlation found in this study could be due to an effect of cortisol on monocyte levels influenced by sex (females). From our results, donkeys differ from other species as the number of monocytes decreases during pregnancy. This pattern has also been found in pigs [39]. During pregnancy in humans and in rats, there is an increase in the number of peripheral blood monocytes, and they may play a role in the pathophysiology of preeclampsia [40]. It might be possible that in jennies, an increase in monocytes poses a risk for pregnancy.

N/L ratio is a good indicator of stress as the ratio increases in stressed individuals [41]. An increase in N/L ratio in donkeys was observed after working [42]. We did not expect to obtain a negative correlation with N/L ratio in pregnant jennies. In most studies of vertebrates, however, in which researchers looked for correlations between baseline measures of cortisol or corticosterone (CORT) and N/L or heterophil to lymphocyte ratios, no correlation was found [43]. In fact, in one study, a weak negative correlation was found [43]. The negative correlation with N/L ratio found in pregnant jennies seems at first paradoxical. At present, however, the reason for this discrepancy as well as probable impacts for pregnancy in donkeys is unclear and is surely a worthwhile focus for future research.

The precise function of basophils has not been determined, although it is known that granules contain heparin, histamine, chemotactic factor of anaphylaxis, and a platelet-activating factor [37]. Basophils are significantly lower in pregnant jennies compared to adults. In humans, corticotrophin-releasing factor (CRF) and adrenocorticotrophic hormone (ACTH) were shown to activate basophils. The basophil response to CRF and ACTH correlated with the serum cortisol concentration in normal controls, but not in patients with idiopathic urticaria [44]. The relationship between cortisol and basophils has not been widely investigated. Unfortunately, we cannot answer whether or not cortisol regulated basophils number in pregnant jennies only because, to the authors’ knowledge, nobody has examined this correlation in pregnant females of other species. Levels of cholesterol and triglycerides are within the values reported by other authors [26,45,46,47]. Levels of cholesterol were significantly higher in juvenile animals compared to in adults, and a similar finding has been reported by other authors [45,47]. Triglyceride levels were significantly lower in the young compared to adults. Similar findings have been reported by other authors [48]. Serum triglyceride levels in donkeys tend to increase with increasing body weight [49]. We found a weak but positive correlation between cortisol and triglycerides in adult females. It could be interesting to evaluate a dynamic relationship between the hormone “widely known as the body’s stress hormone” and triglycerides in donkeys, since they appear to be more predisposed than large-breed horses to suffer from hyperlipemia. 

## 5. Conclusions

The present study showed values of basal cortisol in donkeys. Levels were similar in young and adult animals. In late pregnancy, levels of cortisol in jennies are significantly higher than in the adult group. There was significant positive correlation between cortisol and monocytes and between cortisol and triglycerides in adult females, and between cortisol and monocytes and basophils in pregnant jennies. There was significant negative correlation between cortisol and N/L in pregnant jennies. Cortisol has a variety of effects on different functions throughout the body, which are rarely investigated. It is important to remind the reader that cortisol is not always a reliable indicator of stress or compromised welfare; for instance, horses whose welfare is clearly compromised show low levels of circulating cortisol [50]. Limitations need to be considered when interpreting the substantive results of this study: the data come from a correlational analysis that cannot decisively establish causal direction. Our results could suggest that cortisol does not play the same role across the lifespan but rather serves different functions relating to the modulation of the immune system and lipid metabolism, something that should be further investigated.

## Figures and Tables

**Figure 1 animals-12-00841-f001:**
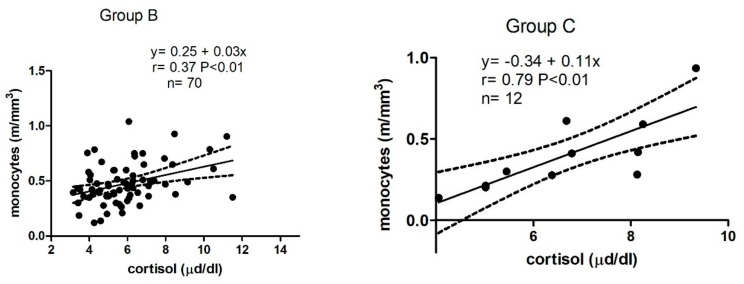
Linear regression model results obtained between serum cortisol and leucocyte profile in Group B (barren jennies) and in Group C (pregnant jennies).

**Figure 2 animals-12-00841-f002:**
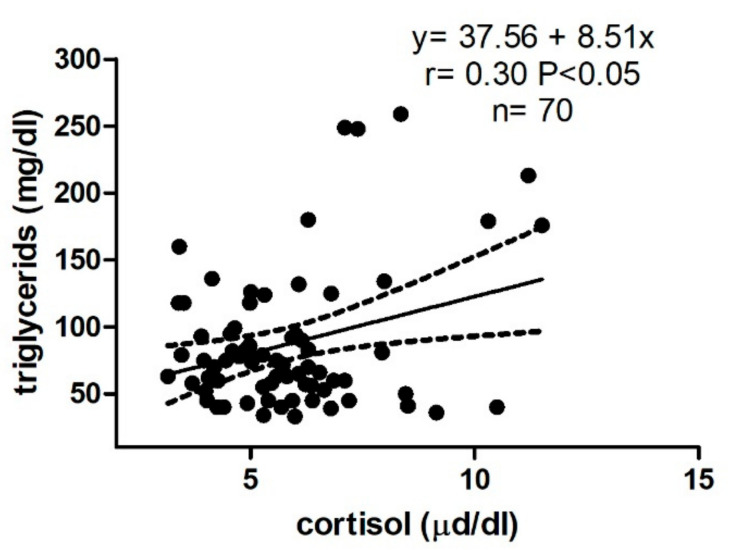
Linear regression model results obtained between serum cortisol and serum triglycerides in Group B (barren jennies).

**Table 1 animals-12-00841-t001:** Baseline values of cortisol (calculated by median, 2.5th–97.5th percentiles) in all donkeys (females and males), in young and adult females (A and B, respectively), and in pregnant jennies (Group C) expressed in µg/dL (Conventional Units). To convert µg/dL to ng/mL values of cortisol must be multiplied by 10. To convert µg/dL to nmol/L (International System of Units) values of cortisol must be multiplied by 27.59.

Analyte	All Donkeys (*n* = 97)	Group A (*n* = 9)	Group B (*n* = 70)	Group C (*n* = 12)
Serum Cortisol (µg/dL)	5.64 (3.40–10.54)	4.67 (4.14–7.65)	5.58 (3.19–10.47)	6.73 * (4.05–9.34)

Mann–Whitney test * *p* < 0.05 vs. Group B.

**Table 2 animals-12-00841-t002:** Mean ± standard deviation of leucocytes in all donkeys (females and males) and in females in relation to age (A and B) and pregnancy (Group C).

White Cell Line	All Donkeys (*n* = 97)	Group A (*n* = 9)	Group B (*n* = 70)	Group C (*n* = 12)	Reference Intervals ^a^
WBC (m/mm^3^)	11.40 ± 2.71	12.59 ± 2.73	11.40 ± 2.83	10.03 ± 1.65	
NEU (m/mm^3^)	4.78 ± 1.52	4.97 ± 0.89	4.70 ± 1.56	4.65 ± 1.09	2.4–6.3
LYM (m/mm^3^)	4.95 ± 2.27	6.36 ± 2.13	4.91 ± 2.43	4.06 ± 1.09	2.2–9.6
MON (m/mm^3^)	0.50 ± 0.20	0.68 ± 0.23	0.47 ± 0.18	0.41 ± 0.23	0–0.75
EOS (m/mm^3^)	1.00 ± 0.53	0.45 ± 0.13 ***	1.13 ± 0.51	0.78 ± 0.40 *	0.1–1.2
BAS (m/mm^3^)	0.08 ± 0.05	0.08 ± 0.04	0.08 ± 0.04	0.05 ± 0.04 **	0–0.13
N/L ratio	1.13 ± 0.52	0.87 ± 0.22	1.13 ± 0.56	1.21 ± 0.35	

WBC: Total leukocyte; NEU: neutrophils; LYM lymphocytes; MON: monocytes; EOS: eosinophils; BAS: basophils. N/L: neutrophil-to-lymphocyte ratio. Mann–Whitney test vs. Group B * *p* < 0.05; ** *p* < 0.01; *** *p* < 0.001 ^a^ Reference intervals from Burden et al. [26].

**Table 3 animals-12-00841-t003:** Mean ± standard deviation of serum cholesterol and triglycerides in all donkeys (females and males) and in jennies in relation to age (A and B) and pregnancy (Group C).

Serum Lipids	All Donkeys (*n* = 97)	Group A (*n* = 9)	Group B (*n* = 70)	Group C (*n* = 12)	Reference Intervals ^a^
CHOL (mg/dL)	73.91 ± 20.80	98.11 ± 32.67 **	71.63 ± 17.53	67.00 ± 19.43	54–112
TRI (mg/dL)	81.73 ± 49.08	55.56 ± 23.40 *	87.60 ± 52.47	77.08 ± 41.83	53–248

Mann–Whitney test vs. Group B * *p* < 0.05; ** *p* < 0.01. ^a^ Reference intervals from Burden et al. [26].

## Data Availability

The data that support the findings of this study are available on request from the corresponding author, [D.A.].

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
