# Peer review of "Serum Cortisol and Its Correlation with Leucocyte Profile and Circulating Lipids in Donkeys (Equus asinus)"

_animals, 2022, doi:10.3390/ani12070841_

Round 1

Reviewer 1 Report

The paper submitted by Alberghina et al to “Animals” reports serum cortisol levels and their correlation with leucocyte profiles in donkeys. In their small, explorative experiment, the authors analysed single samples from young and adult animals (some also from pregnant females).

I agree that compared to horses, only a few papers dealing with cortisol in donkeys have been published. However, I was surprised to find several uncited papers via a quick and simple literature search (“donkey” AND “cortisol”; see refs below), which to my opinion should be included in the introduction and/or discussion.

As the authors correctly point out, reference values need to be established by measuring many samples. Thus, I fear sample number is too low here (especially for males) to be able to give “reference intervals” (thus I also suggest to change the title). Besides, especially for cortisol a diurnal rhythm may exist (so more samples per animal would be needed), and many other factors are known to influence its level (e.g. also episodic fluctuations). In conclusion, the given concentrations can only be regarded as a pilot study.

Reference values are also depending upon the analytical method – no details are given here, but those need to be added (see below).

Besides, I fear that the conclusions drawn in this paper are not valid ones. You only have single, point in time cortisol measurements (from different individuals), so it is impossible to determine any effects on other parameters (especially ones taken at the same time). Besides, correlations should never be taken as causations!

Below please find further detailed comments (ordered by appearance in the ms):

My suggestion for a modified title: “Serum Cortisol and Leucocyte Profile in Donkeys (Equus Asinus)”

Line 14: Reference values are not defined!

Line 15: “… age did not influence… state increased…”

Line 15/16: I don’t think that the experimental design allows aiming at this. You only analysed single samples from different individuals. And different parameters measured from the same sample cannot influence each other (cave: correlation is not causation!).

Line 18: “functions (plural)… change!

Line 27 (and elsewhere). The dimension “µg/dl” is a very unusual one for animal studies. I suggest using “ng/ml” instead.

Line 32: “cortisol has”

Line 37: “… secreted by the…”

Line 63: replace “gender” with “sex” – you are talking here about the biological term (not the social one).

Line 66: “veterinary care”

Line 75: Replace “interactions” with “correlations”.

Line 85/86: Especially the number of males is very low!

Line 101: You aim at giving reference values and those are method dependent. Therefore it is important to give more details of the method here (cross-reactions of the antibody, sensitivity, validity, etc….)

Line 115: “.. study had”

Line 118/119: I think you cannot compare pregnant mares with adults in general, but only adult females (especially as you have a low number of males; I suggest to remove those when comparing groups B and C)

Table 1: Why do you give mean and standard deviations, when groups were not normally distributed?

Line 123: “Correlations” instead of “Interactions”

Figure 1: I fear the one outlier (>10µg/dl) will have affected correlations dramatically (what about results when excluding it?)

There is a lot of tables and figures showing correlations, but I question the usefulness (especially as there seems to be no overall patterns – so only arbitrary).

Line 163: Cortisol levels… were…

Line 170: This is not a valid conclusion. Cortisol levels cannot influence leucocyte parameters from the same sample.

Line 207/208: That is not only here impossible (see above)

Line 209: …authors’

Line 212: Reference values are not shown.

Above cited references:

Jiang, GM., Zhang, XH., Gao, WP., Ji, CL., Wang, YT., Feng, PX., Feng, YL., Zhang, ZP., Li, L., Zhao, FW. (2021): Transport stress affects the fecal microbiota in healthy donkeys. J. Vet. Internal Med. 35, 2449-2457. https://doi.org/10.1111/jvim.16235

Zhao, FW., Jiang, GM., Ji, CL., Zhang, ZP., Gao, WP., Feng, PX., Li, HJ., Li, M., Liu, HB., Liu, GQ., Magalhaes, HB., Li, JJ. (2020): Effects of long-distance transportation on blood constituents and composition of the nasal microbiota in healthy donkeys. BMC Vet. Res. 16, 338. https://doi.org/10.1186/s12917-020-02563-5

Bonelli, F., Rota, A., Aurich, C., Ille, N., Camillo, F., Panzani, D., Sgorbini, M. (2019): Determination of salivary cortisol in donkey stallions. J. Equine Vet. Sci. 77, 68-71. https://doi.org/10.1016/j.jevs.2019.02.027

Fazio, E., Fragala, S., Fragala, Santo., Ferlazzo, A.,  Cravana, C., Torrisi, K., Medica, P. (2017): Progesterone, estradiol-17 beta, cortisol, and hematological profile during the estrous cycle of lactating jennies: Preliminary and comparative observations. J. Equine Vet. Sci. 56, 26-34. https://doi.org/10.1016/j.jevs.2017.04.010

Forhead, AJ., Smart, D., Smith, RF., Dobson, H. (1995): Transport-induced stress responses in fed and fasted donkeys. Res. Vet. Sci. 58, 144-151. https://doi.org/10.1016/0034-5288(95)90068-3

Author Response

Dear Reviewer,

Thank you for your careful reading of our text. We have addressed many of your concerns with as much detail as possible and, where we agree with your comments, we have revised our manuscript accordingly.  We are including your helpful suggestions and have clarified the text when needed. We are confident that the new version of the manuscript will be greatly improved.

Please, find below your comments (black font) and our responses (blue font) inserted after each comment.

Looking forward hearing from you soon.

Best regards,

Daniela Alberghina

Reviewer 2 Report

This study adds new scientific data for the hematology in animals. Authors effort is  appreciated.

Please give the scientific name of horse in the text when introducing the term horse.

In line 96 , Eosinophils is abbreviated as EOS, but in the text it is used as EO. Please correct this

Line 121- *P<0.05 vs group B this is not clear. Did you analyse the data of all

three groups? better to give actual P value in the table and use superscripts to shoe the difference.

Line 159- please rewrite the sentence in clearly mentioning that the Observed values of Donkeys were overlapped with the values of horse.

In general better to develop the discussion section explaining few more examples for the observed values.

Author Response

(The authors gave the same response as above.)

Reviewer 3 Report

Dear authors,

I read with interest your manuscript and I found merit in it. However, I have few suggestions that in my opinion may be helpful to improve the text and description. 

The structure of the introduction is, to me, unbalanced. That means that it is not structured in a way to adequately introduce the reader to the point of your investigation. Specifically,  I would invite you to reconsider the order and length of each argument you deal with, but first please, start with this order: 1-brief physiological description of stress and cortisol in animals and how it can be assessed. 2- Implications for the determining the circulating cortisol for the practice; 3- rationale to use this determination in donkeys; 4- how to interprete biochemical results (this is the gap in the literature); 5-your hypothesis and how you meant to fill the gap (you used different animal categories and breeds, please provide a potential role for this design right at the end of the Intro, before you start with material and methods) so clearly state your objectives. 

M&M: are fine, though I would invite authors to improve the description of farming conditions. You report 2 breeding farms (are those stud stations? Because the number of animals is high if considering the normal number of donkeys raised per farm). So please, describe the type of farming conditions which is of basic importance for the background and evaluation of the soundness or results. 

Additionally, feeding is also totally missing. Please add.

Cortisol without biochemistry is unusual. In the introduction you report that cortisol has modulatory effects on several circulating parameters, but none of those were explored. It is also arguable to define the baseline levels without additional metabolic background.

Results. Tables could be better displayed and please report explanation to acronyms used in tables as footnotes. In addition, as to reference intervals of Italian breeds, I would suggest to report values reported by Caldin et al. 2005, Comp Clin Pathol. doi 10.1007/s00580-005-0544-8, reporting on Ragusana donkeys with intervals of reference. Thanks. 

Discussion. It is well developed. I would only invite authors to discuss results on the importance of carotenoids in face of cortisol. As donkeys were fed on hay (in which carotenoids decrease due to photolability in hay if compared to pasture, would it be possible to have different levels of circulating baseline cortisol following different feeding regimens beyond stressing factors? I would warmly invite authors to compare with Cappai et al. 2017, Ecology and Evolution doi 10.1002/ece3.2613 the effect of seasonal variations of carotenoids circulating blood serum of pasture donkeys.

Conclusion. Please, organize conclusion in a way to provide a reply to your hypothesis.

Author Response

(The authors gave the same response as above.)

Round 2

Reviewer 1 Report

Thanks to the author for improving their paper significantly and for making it easy to follow their changes (both in the doc, as well as in the letter). However, there are still several points (please see below), which need to be clarified before the paper can be accepted.

General:

You stated that you deleted “reference” values throughout the text, but it is still present some times (e.g. line 23; 144). Please remove and check once again carefully. As you indicated in your letter, sample number is too low to give robust “reference” values. Besides, those are method dependent, and as you cannot give any details of the method utilized to measure cortisol (and an immunoassay is not specific, though prone to cross-reactions), I am strictly again labelling them “reference” values.

I welcome excluding the few male donkeys from this study, but fear that their values are still included in some paragraphs. Please remove those throughout (e.g.: Tables in the group: “All donkeys”)

Re the dimension: µg/dl – I’m surprised to read that this is the conventional unit; whose convention? In humans, yes (for historical reasons because of old reference values), but in animals? Ok, but I suggest to also indicate that you would need to multiply with 10 to get the frequently used unit: ng/ml) – for sure nmol/L is the IU.

Below please find my detailed comments (ordered by appearance in the ms):

Title: I suggest deleting “values” (besides, “values” – plural won’t match with “its” – singular)

Line 14: add “values” here (“basal cortisol values”), because basal cortisol is not something different than e.g. peak cortisol – it’s always the same molecule.

Line 33: Well, “hypercortisolism” is a strong word; I suggest to reword: “.. higher cortisol values are also found in this species”)

Line 79: “basal levels” are not defined, they may be “evaluated”.

Line 123: “cholesterol” – not capitalized.

Line 134: “examinated”?

Line 169: “triglycerids”

Line 172: ….showed significantly higher CHOL levels, …lower TRI levels…

Line 251: “Levels of …were..”

Author Response

Dear Reviewer,

Thank you for your careful reading of our text. We have revised our manuscript according to your concerns.  We are including your helpful suggestions and have clarified the text when needed. We are confident that the new version of the manuscript will be greatly improved.

Please, find below your comments (black font) and our responses (blue font) inserted after each comment.

Looking forward hearing from you soon.

Best regards,

Daniela Alberghina
